# Nexus between Sustainability Reporting and Firm Performance: Considering Industry Groups, Accounting, and Market Measures

Banu Dincer [1], Ayşe İrem Keskin [2] and Caner Dincer [1,*]

1   Department of Business Administration, Faculty of Economic and Administrative Sciences, Galatasaray University, Çıragan Cad. No: 36, Ortaköy, Istanbul 34349, Turkey
2   Department of Business Administration, Faculty of Economics, Administrative and Social Sciences, Kadir Has University, Cibali Mah. Fatih, Istanbul 34083, Turkey
*   Correspondence: cdincer@gsu.edu.tr; Tel.: +90-212-227-44-80 (ext. 525)

**Abstract:** The relationship between Sustainability Reporting and corporate financial performance is overlapping and multifaceted and it has been an interesting issue for both academics and professionals since the beginning of the millennium. Studies have found divergent results on this relation and the industrial differences are omitted in many papers. Moreover, studies considering developing countries are scarce. The purpose of this study is to shed light on the relationship between sustainability reporting and firm performance in a developing country context. The impact of sustainability reporting is investigated using pooled ordinary least square (OLS) method for panel data regression through two models based on Tobin's Q and ROA. A total of 920 observations for 46 companies with 3 different impact levels based on their environmental effect and 5-year quarterly panel data between 2016–2020. The research used data from Borsa Istanbul (Istanbul Stock Exchange) and also independent variables such as leverage, risk, size, current ratio, growth, sustainability reporting, and the environmental impact level of companies. The results showed that sustainability reporting has a significant positive impact on financial performance according to the ROA model, and a significant negative correlation between risk and financial performance according to both ROA and Tobin's Q models. Considering the environmental impact of companies, the results also reveal a positive relationship between high impact companies' sustainability reporting and short-term financial performance as ROA is an accounting-oriented measure that reveals the company's short-term financial performance. Further research should investigate the impact of sustainability reporting in different markets based on the impact level of companies and the development degree of countries.

**Keywords:** sustainability reporting; financial performance; Tobin's Q; ROA; sustainability impact

## 1. Introduction

Sustainability reporting (SR) is one of the prominent research areas which has received exponentially increasing attention in recent years. SR covers environmental, social, and governance (ESG) issues and sustainability concerns that stakeholders demand from organizations to manage their risks and opportunities. To ensure accountability and transparency, there is a tendency to create a new global system for SR. In 2021 and 2022, tremendous advances have been realized concerning regulations and standards. In November 2021, the IFRS (International Financial Reporting Standards) Foundation Trustees released the establishment of the International Sustainability Standards Board (ISSB) to prepare a global sustainability-related standard. Meanwhile, the European Council in December 2022 accepted the Corporate Sustainability Reporting Directive (CSRD) which generated the release of European Sustainability Reporting Standards (ESRS) by the European Financial Reporting Advisory Group (EFRAG). The latter means that approximately 50,000 companies must disclose data according to ESRS, which will start applying between 2024 and 2028.

With this fast evolution of the SR landscape, it is expected that this prevalently discussed topic will continue to be discussed as it has no consistent conclusions about its impact on corporate financial performance [1].

Sustainable reporting can be defined as the measurement, disclosure, and accountability of organizational performance in achieving sustainable development goals to internal and external stakeholders [2]. Thus, SR can reduce the information asymmetry and increase the transparency of the company's sustainability activities and incite investors to direct their investments to companies with positive impacts. Moreover, SR gives a competitive advantage to the companies, in their market or industry [3]. Considering these advantages, companies try to profit from SR and publish their reports. However, the studies in the field also report an insignificant or inverse relationship between SR and financial performance. So, some studies report an increased financial performance [4–7], albeit others state an inverse [8,9] or an inconclusive relationship between them [10–12]. Ref. [13] affirmed on the impact of SR on financial performance that most of the studies pointed to a positive relationship between SR and financial performance. However, due to the mixed results, ref. [13] also recommended further research may yield more consistent findings. Thus, researchers have noticed that consequent to these different findings, sectoral analysis is scarce in SR [14,15]. Indeed, as the ESG factors vary from one sector to the other, analyzing the relationship between SR and financial performance without categorizing the sectors may be the reason behind these mixed results [16,17]. So, these studies with these divergent results lack a sectorial approach to sustainability reporting [18]. The sectorial differences, the development stage of the market in the study, and the measurement choices shape the impact of SR. Although many studies have considered the impact of SR from a holistic point of view [14], scant attention has been paid to sectorial differences on this topic.

It affirmed that the political, social, and economic characteristics of developing countries affect the SR approach of the companies in these countries [19]. Moreover, most of the world's population lives in developing countries. Therefore, this study aims to elucidate the relationship between SR and firm performance in a developing country context. This paper synthesizes recent studies to use three sectorial levels (high, medium, and low impact sectors) and two different measures. This approach considers the accounting and market measures that show the short-term and long-term impact of SR on performance and the effect of the firm's industry's impact level on the environment.

The gap in the literature stems from the divergent results, the consideration of sectorial differences, and the developing country context. Accordingly, this study is expected to add to the literature and guide further research on this topic, especially with the two models of performance and classification of firms.

This study has several contributions to the literature. Firstly, it posits the impact of companies' SR practices on the market-oriented as well as accounting-oriented measures, respectively, for long-term and short-term financial performance, specifically in a developing country. Secondly, although previous studies have made cross-sectoral analyses [20], this research categorizes sectors in terms of their environmental effects consisting of three categories covering industry groups. Finally, the results aim to broaden the insight into SR implications for a firm's financial performance and shed light on sectoral divergence, which should help the stakeholders understand the meaning and the necessity of recently mandated SR.

This paper is presented as follows: we will review the relevant literature and describe the hypothesis, the data used, and the research methodology. The conclusion and the recommendations for future research are discussed in the final section.

## 2. Literature Review

The terms SR and ESG are used interchangeably and in an overlapping manner in the literature [21]. Some studies assess the link between financial performance and ESG factors [22], and some others fulfill this aim by using sustainability reports [23,24]. However, this is not entirely accurate. It must be emphasized that SR refers to the information

that companies provide about their performance to the outside world on a regular basis in a structured way. Through sustainability reporting, companies communicate their performance and impact on a wide range of sustainability topics, spanning environmental, social, and governance parameters. ESG reports on the other side are reporting frameworks, disclosing environmental, social, and corporate governance data and they can be included in the Sustainability Reports.

According to the stakeholder theory, companies need to fulfill the expectations of diverse stakeholders, not only by disclosing financial, but also non-financial information. Hence, SR by providing transparency and accountability enables stakeholders to make informed and conscious decisions. In the meantime, organizations can identify where they are not meeting societal expectations and can take steps to solve these issues, which are in line with the legitimacy theory [25] and the stakeholder theory [26,27]. Therefore, from the perspective of stakeholder theory, companies can highlight their reputation, gain the support of the stakeholders, and attract investments, which lead to better financial performance [28–30]. The demonstration of the commitment to sustainability and building trust with stakeholders and thus, with society, will affect the financial success in the long term and create value, as legitimacy is vital for the long-run prosperity of the company [31,32].

In this line of research, mixed results are obtained based on accounting measures as well as market measures. Return on Assets (ROA) is widely used in numerous studies to measure the accounting aspects, and their relationship with SR disclosed, respectively, a positive relationship in some studies [33–35], a negative in some others [36], or insignificance [37,38]. Market performance is measured in many others with Tobins' Q [39–42] to assure the accountability and transparency of the firm value. Table 1 resumes the recent studies about SR and firm performance.

**Table 1.** Recent literature review of SR, ESG factors, and financial performance.

| Article | Subject | Focus | Model Used | Results |
|---|---|---|---|---|
| Mattera et al., 2022 [43] | Implementation of sustainable business models and its effect on firm's performance | FTSEMIB Index Companies' financial performance during COVID-19 in the year 2020 | Chi-square and correlation analyses of the share price | Positive association between sustainable strategies and firm's financial performance |
| Oware, K.M., Mallikarjunappa, T. 2022 [44] | Examination of the moderating effect of mandatory CSR reporting on financial performance of listed firms in India | Indian stock market companies for 800 firm-year observations from 2010 to 2019 | Hierarchical regression and panel regression with fixed effect assumptions | Positive relation between financial performance (ROA and Tobin's Q) and CSR expenditure |
| Thomas, C.J., et al., 2021 [22] | Empirical analysis of sustainability practices on firm performance using ESG data | Malaysia stock market companies for 36 public listed firms reporting ESG scores from 2015 to 2019 | Static panel regression | Positive relation between ESG and financial performance, ROA, ROE, and Tobin's Q, but only significant for ROE |
| Buallay, A. et.al., 2021 [45] | Research on the relationship between the level of sustainability reporting and firm's performance | 20 different smart city companies for 3536 observations from 2008 to 2017 | Multiple regression model | Positive significant association between ESG and ROA, ROE; negative significant association between ESG and Tobin's Q |

**Table 1.** *Cont.*

| Article | Subject | Focus | Model Used | Results |
|---|---|---|---|---|
| Pham, D. C. et al. (2021) [46] | Sustainability practices on the financial performance | Swedish companies for 116 listed firms in the year 2019 | Multivariate regression model | Positive relationship between corporate sustainability and performance |
| Buallay, A., et al., 2021 [47] | Research on the relationship between ESG and a bank's performance (Tobin's Q) | Stock exchanges of MENA countries for 59 listed banks from 2008 to 2017 | Fixed-effect regression model and IV-GMM (generalized model of moments) | Positive impact of ESG on performance; social performance plays a negative role in determining a bank's profitability and value |
| Buallay, A. 2019 [48] | Research on sustainability reporting's effect on performance with a comparison between manufacturing and banking sectors | Companies in 80 countries (932 manufacturers and 530 banks) for 11,705 observations from 2008 to 2017 | Pooled data regression Model | Positive impact of ESG on performance in the manufacturing sector, besides negative effects in the banking sector |

As shown in Table 1, recent studies use different measures on different markets. Although these studies have found mainly positive relationships between SR and firm performance indicators, previous studies have found insignificant and negative relationships and their focus is on developed markets. These recent studies suggest that managers should allocate a proportion of their resources towards reporting on their attempts to mitigate the harmful impacts of their business operations, especially those in high-impact industries whose operations could be remarkably destructive. Accordingly, firms are classified as high, medium, or low impact based on their environmental effect according to FTSE in our study. Moreover, our focus is on developing market firms and analysis is based on market and accounting measures.

## 3. Materials and Methods

XU100 and XUSRD are indices in Borsa Istanbul (BIST). BIST indices the companies that report their SR activities on BIST Sustainability Index (XUSRD). BIST 100 Index (XU100) is a capitalization-weighted index that tracks the financial performance of 100 primary companies chosen from the National Market. Accordingly, all firms in the sample are selected from XU100 and XUSRD indices to have accurate and detailed data.

The sample used in the research consists of a total of 46 firms all included in XU100. As the financial firms have different approaches in their financial statements, they are excluded from the analysis to prevent any bias. So, the firms used in the analysis are all in XU100 and non-financial. Thirty-one of them are publishing Sustainability Reports and they have been in the BIST Sustainability Index (XUSRD) for the whole 5-year period uninterruptedly. Fifteen others are also XU100 firms, but they do not practice SR or have continuously published Sustainability Reports for the research period. We finally have 920 observations for 46 companies and 5-year quarterly panel data between 2016–2020.

Our sample has 9 sectors and 15 industry groups. We use Standard and Poor's Global Industry Classification Standard (GICS) to determine the sectors and industry groups, combine the data of the sector and industry group from Wharton Research Data Services, Compustat Global Database, and collect all financial data from Bloomberg Database. Similar databases have been widely applied in the literature by researchers and academicians examining the relation between SR and financial performance (e.g., [49–53]). Table 2 shows sample company classification by sector.

**Table 2.** Company classification by sector.

| Sector | Number | Percentage |
|---|---|---|
| Industrials | 12 | 27 |
| Consumer Staples | 7 | 15 |
| Health Care | 1 | 2 |
| Consumer Discretionary | 11 | 24 |
| Materials | 7 | 15 |
| Energy | 2 | 4 |
| Utilities | 3 | 7 |
| Information Technology | 1 | 2 |
| Communication Services | 2 | 4 |
| Total | 46 | 100 |

Table 3 shows company classification by industry group.

**Table 3.** Company classification by industry group.

| Industry Group | Number | Percentage |
|---|---|---|
| Capital Goods | 10 | 22 |
| Food and Staples Retailing | 2 | 4 |
| Pharmaceuticals, Biotechnology, and Life Sciences | 1 | 2 |
| Automobiles and Components | 6 | 13 |
| Household and Personal Products | 1 | 2 |
| Materials | 7 | 16 |
| Energy | 2 | 4 |
| Consumer Services | 1 | 2 |
| Utilities | 3 | 7 |
| Food, Beverage, and Tobacco | 4 | 9 |
| Consumer Durables and Apparel | 3 | 7 |
| Retailing | 1 | 2 |
| Technology Hardware and Equipment | 1 | 2 |
| Transportation | 2 | 4 |
| Telecommunication Services | 2 | 4 |
| Total | 46 | 100 |

Sample companies are classified as high, medium, or low impact based on their environmental effect according to FTSE (Financial Times Stock Exchange) Russell, which is in the London Stock Exchange Group. The FTSE sector classification is in Table 4.

Our sample has 30 companies with high impact, 13 with medium impact, and 3 with low impact on the environment. Of the sample, 18 high-impact companies, 10 medium-impact companies, and 3 low-impact companies have corporate sustainability reporting. Figure 1 shows the sector impact percentages of companies in the total sample, with 65% high impact and 28% medium impact sector.

Table 4. Sector Classification.

| High Impact Sectors | Medium Impact Sectors | Low Impact Sectors |
|---|---|---|
| Agriculture<br>Airports<br>Building Materials (includes Quarrying)<br>Chemicals and Pharmaceuticals<br>Construction<br>Major Systems Engineering<br>Fast Food Chains<br>Food, Beverages, and Tobacco<br>Forestry and Paper<br>Mining and Metals<br>Oil and Gas<br>Power Generation<br>Road Distribution and Shipping<br>Supermarkets<br>Vehicle Manufacture<br>Waste<br>Water<br>Pest Control | DIY and Building Supplies<br>Electronic and Electrical equipment<br>Energy and Fuel Distribution<br>Engineering and Machinery<br>Financials not elsewhere classified (see right)<br>Hotels, Catering, and Facilities Management<br>Manufacturers not elsewhere classified<br>Ports<br>Printing and Newspaper Publishing<br>Property Developers<br>Retailers not elsewhere classified<br>Vehicle Hire<br>Public Transport | Information Technology<br>Media<br>Consumer/Mortgage Finance<br>Property Investors<br>Research and Development<br>Leisure not elsewhere classified (Gyms and Gaming)<br>Support Services<br>Telecoms<br>Wholesale Distribution |

Source: FTSE4Good Index Series Inclusion Criteria, 2010, p.3. (Source: https://blog.metu.edu.tr/sascigil/files/2016/02/FTSE4Good_Inclusion_Criteria.pdf accessed on 20 October 2022).

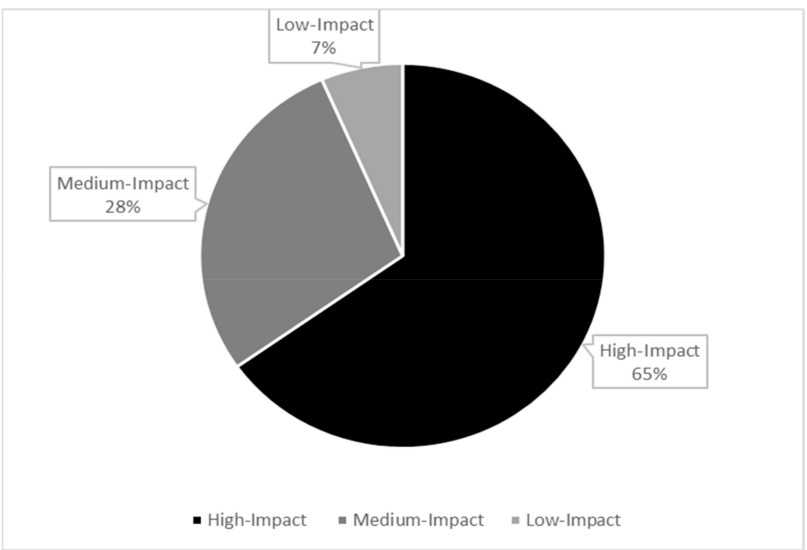

**Figure 1.** Sector impact percentages.

Table 5 presents the variables of our two models. The dependent variables are Tobin's Q and ROA. In Model 1, Tobin's Q is a market-oriented measure for long-term financial performance [54,55], whereas ROA is an accounting-oriented measure that reveals the company's short-term financial performance [56] in Model 2. The independent variables include leverage, risk, size, current ratio, growth, SR, high impact, medium impact, and low impact. SR is a dummy variable that takes the value 1 when the company is in XUSRD and 0 otherwise. These are variables used in similar studies in the literature to measure different aspects of performance (e.g., [57,58]). Moreover, high impact, medium impact, and low impact are dummy variables that have the value 1 when they belong to high, medium, and low effects on the environment, and others are 0. All variables were winsorized with 0.1 value to trim outliers. We use the pooled ordinary least square (OLS) method for panel data regression using Stata 15 for the analysis to estimate Models 1 and 2, testing the relationship between SR and financial performance [59]. As the results of the relationships between variables are mixed in the literature based on both accounting

measures (ROA model) and market measures (Tobins' Q model). The study aims to analyze the relationship between sustainability reporting and firm performance, using two equations based on accounting and market dimensions of performance. The models are constructed to encapsulate the effects of sustainability reporting on firm performance. They are as follows:

**Table 5.** Variables.

| Variable | Definition |
|---|---|
| Tobin's Q * | Total market value of the stock/total asset value of firm-measure of long-term financial performance, market-oriented |
| ROA | Return on Asset—net income/total asset—measure of short-term financial performance, accounting-oriented |
| Leverage | Total liabilities/total assets |
| Risk | Natural logarithm of (total debt/total asset) |
| Size | Natural logarithm of total assets |
| Current Ratio | Current asset/current liabilities |
| Growth | Natural logarithm of yearly sales growth |
| SR | Sustainability reporting |
| High Impact | Companies classified in sectors that have a high impact on the environment |
| Medium Impact | Companies classified in sectors that have a medium impact on the environment |
| Low Impact | Companies classified in sectors that have a low impact on the environment |

\* (Bloomberg definition: Tobin's Q compares the total value of the prices of stocks with the cost of replacing the underlying assets of those same stocks or corporate net worth. It is argued that when the stock market trades at a discount to the replacement cost of its assets, the market is relatively inexpensive. (Source: https://www.bloomberg.com/professional/blog/insight-tobins-q-implies-overvalued-stock-market/#:~:text=The%20Tobin%20q%20compares%20the,the%20market%20is%20relatively%20inexpensive. accessed on 1 November 2022).

(1) Tobin's Q = $\beta_0 + \beta_1$ Risk$_{i,t}$ + $\beta_2$ Leverage$_{,t}$ + $\beta_3$ Size$_{i,t}$ + $\beta_4$ Current Ratio$_{i,t}$ + $\beta_5$ Growth$_{i,t}$ + $\beta_6$ High Impact$_{i,t}$ + $\beta_7$ Medium Impact$_{i,t}$ + $\beta_8$ SR$_{i,t}$ + $\varepsilon_{i,t}$;

(2) ROA = $\beta_0 + \beta_1$ Risk$_{i,t}$ + $\beta_2$ Leverage$_{,t}$ + $\beta_3$ Size$_{i,t}$ + $\beta_4$ Current Ratio$_{i,t}$ + $\beta_5$ Growth$_{i,t}$ + $\beta_6$ High Impact$_{i,t}$ + $\beta_7$ Medium Impact$_{i,t}$ + $\beta 8$ SR$_{i,t}$ + $\varepsilon_{i,t}$.

In these models, we used Robust standard errors to consider heteroscedasticity and autocorrelation in the panel dataset. The reason for robust standard errors in panel data is that the idiosyncratic errors can have heteroskedasticity or autocorrelation, or both. Robust standard errors account for heteroskedasticity in a model's unexplained variation. That is, if the amount of variation in the outcome variable is correlated with the explanatory variables, robust standard errors can take this correlation into account to obtain unbiased standard errors of OLS coefficients. [45,60,61]. A length of 5 years is used in the analysis and the results turn out to be quite robust to changes in the selected length.

## 4. Results

First, we analyzed the descriptive statistics of panel data as presented in Table 6. There were no missing values in the panel data. According to our 920 observations, the data show that ROA has the lowest mean (0.57), while Tobin's Q has (1.34) as the mean value and size has the highest value (15.82).

The mean variance inflation factor (VIF) is below 5 as shown in Table 7, so there is no multicollinearity in the model. Low-impact sectors were omitted because of collinearity in the regression analysis. We used robust regression because heteroscedasticity is present according to Breusch–Pagan/Cook–Weisberg test.

The results displayed in Table 8 show that Model 1 and Model 2 have a high explanatory power and statistical significance as they have F-tests with *p* values less than 5%. The results also reveal that the independent variables explain 27% of the variation of ROA and, respectively, 27.2% of the variation of Tobin's Q according to the R-squared value.

**Table 6.** Descriptive statistics.

| Variable | Obs | Mean | Std. Dev. | Min | Max |
|---|---|---|---|---|---|
| Tobin's Q | 920 | 1.34 | 0.407 | 0.895 | 2.18 |
| Risk | 920 | 2.978 | 1.199 | 0 | 4.042 |
| Leverage | 920 | 0.578 | 0.208 | 0.188 | 0.841 |
| Size | 920 | 15.822 | 1.158 | 13.971 | 17.625 |
| Current Ratio | 920 | 1.646 | 0.841 | 0.753 | 3.465 |
| Growth | 920 | 2.091 | 2.224 | −2.511 | 4.128 |
| ROA | 920 | 0.057 | 0.055 | −0.021 | 0.15 |

**Table 7.** The mean variance inflation factor and VIF Values.

| Variable | VIF | 1/VIF |
|---|---|---|
| Risk | 2.680 | 0.373 |
| Leverage | 4.520 | 0.221 |
| Size | 1.510 | 0.660 |
| Current Ratio | 3.620 | 0.276 |
| Growth | 1.010 | 0.987 |
| High Impact | 4.040 | 0.248 |
| Medium Impact | 3.900 | 0.257 |
| SR | 2.240 | 0.447 |
| Mean VIF | 2.940 | |

**Table 8.** Summary of regression analysis for panel data.

| | (Model 1) | (Model 2) |
|---|---|---|
| **VARIABLES** | **Tobin's Q** | **ROA** |
| Risk | −0.0914 *** | −0.0141 *** |
| | (0.0189) | (0.00238) |
| Leverage | 1.219 *** | 0.0160 |
| | (0.119) | (0.0147) |
| Size | −0.117 *** | −0.00411 *** |
| | (0.0121) | (0.00152) |
| Current Ratio | 0.169 *** | 0.0187 *** |
| | (0.0287) | (0.00335) |
| Growth | 0.0190 *** | 0.00353 *** |
| | (0.00497) | (0.000698) |
| High Impact | 0.0850* | 0.0236 *** |
| | (0.0435) | (0.00621) |
| Medium Impact | −0.145 *** | 0.00643 |
| | (0.0426) | (0.00633) |
| SR | 0.0610 | 0.0281 *** |
| | (0.0407) | (0.00509) |
| Constant | 2.379 *** | 0.0804 *** |
| | (0.237) | (0.0282) |
| Observations | 920 | 920 |
| F-test | 60.330 | 46.853 |
| Sig. | 0.000 | 0.000 |
| R-squared | 0.272 | 0.270 |

Robust standard errors in parentheses *** $p < 0.01$

Tobin's Q Model shows that risk ($\beta = -0.0914$), size ($\beta = -0.117$), and medium impact ($\beta = -0.145$) have a negative significant effect on Tobin's Q, while leverage ($\beta = 1.219$), current ratio ($\beta = 0.169$), growth ($\beta = 0.0190$), high impact ($\beta = 0.0850$) have a positive significant effect on Tobin's Q. Accordingly, the model indicates SR does not influence the long-term financial performance.

ROA Model indicates that risk ($\beta = -0.0141$) and size ($\beta = -0.00411$) have a negative significant effect on ROA, while current ratio ($\beta = 0.0187$), growth ($\beta = 0.00353$), high impact ($\beta = 0.0236$), and SR ($\beta = 0.0281$) have a positive significant effect on ROA. Moreover, the model shows that leverage and medium impact do not influence the company's short-term financial performance.

## 5. Discussion

The purpose of the study is to analyze the relationship between SR and financial performance, taking into consideration the market-oriented and accounting-oriented measures that enable to have a long-term and short-term perspective in a developing country context. Although this method has been used in other similar studies using cross-sectoral analysis (e.g., [20]), this research categorized sectors into 3 groups in terms of their environmental effects. The analysis showed that market-oriented and accounting-oriented data both have an impact on financial performance. The SR showed a significant positive impact on ROA, a short-term oriented measure.

We used the data from listed companies in XU100 and XUSRD indices in Borsa Istanbul (BIST) as a research sample making a total of 920 observations for 46 companies and 5-year quarterly panel data between 2016–2020, covering 9 sectors and 15 industry groups.

Based on our empirical analysis, SR has a significant positive impact on financial performance according to ROA, the accounting-oriented model, although this relation seems to be conflicting, as suggested by Buallay (2021). SR and related activities can help companies build trust-based relationships with consumers and enhance corporate reputation to improve financial performance. The model also indicates SR does not influence long-term market performance [46,62].

In addition, there is a significant negative correlation between risk and financial performance according to both two ROA and Tobin's Q models. Risk is related to the total obligations of the company. Accordingly, an increase in risk makes it difficult to reach credit with low interest and causes additional financial costs, decreasing ROA [48,63]. Concerning Tobin's Q, we can affirm that the higher risk of a company affects market perception negatively.

Our sample is composed mainly of industrial and consumer discretionary sectors. In these sectors, it is not easy to reflect asset investments' effect on profit in the short term. Thus, ROA and size are negatively related [22]. Accordingly, the market also responds negatively to the size as even in the long term; it is difficult to reach higher profits from an investment made by large companies. The companies in the sample are already at a certain size; therefore, investors do not expect extraordinary returns from total asset investments causing a negative relationship between size and Tobin's Q [64].

Current ratio is an essential indicator of liquidity, and the consequently higher current ratio has a significant positive impact on ROA and Tobin's Q [22]. Current ratio variations [22] and growth [64] affect Tobin's Q in the same direction.

Considering the environmental impact of companies, the results reveal a positive relationship between higher impact and ROA as most companies in our sample report sustainability. Corporate sustainability reporting enhances the company's reputation, improves stakeholders' perception, and strengthens its position in the market to be more profitable [24,39,46,62,64]. The ROA model results encourage companies with sustainability-based organizational structures to have a higher short-term financial performance than the others [48].

High impact sector gives much more detailed information about the environmental issues such as the methods used for manufacturing or the minimum damage they are causing to the environment, for examples, whereas the medium-impact sector highlights primarily the social and governmental issues emphasizing gender equality, human rights, transparency, and accountability. Hence, this study reports that the stakeholders value high-impact sector SR in the short term and the long term significantly in terms of ROA and Tobin's Q, which is in line with stakeholder and legitimacy theory, albeit the medium

impact sector's SR efforts are not considered as such by stakeholders. Environmental investments in medium-effect sectors do not convince investors of their necessity, unlike high-impact sectors.

Creditors and investors perceive high financial leverage as a sign of companies' profit as it is used to amplify returns from an investment. Hence, demand for stock and stock price will increase [22]. However, the leverage level has no significant effect on ROA in the short-term as the investment generally produces an outcome in the long-term; therefore, it has a significant positive effect on Tobin's Q.

Companies pursuing SR are well aware that SR facilitates their efforts in building and protecting their corporate reputation, satisfying their stakeholders through the SR while generating positive outcomes in diverse financial performance objectives. Even in a developing country market, where investors are more short-term oriented, SR has many positive outcomes in our work as companies legitimize their SR activities and affect the expectations of various stakeholders via SR [65]. This is because when a company is committed to SR, it strengthens its reputation and gains the trust of the stakeholders while maintaining operational, financial, and market performance. However, differently, in Tobin's Q model, the SR variable has an insignificant effect, but the leverage has a positive impact. The effect of leverage comes from the positive expectations of market investors.

Our study shows that companies which practice SR have higher values of financial performance in the short term, in line with works such as [4]. According to the results, the current ratio, growth, and the impact degree of the companies have a significant positive impact both in the short and long term. SR is more considered in the short term and leverage in the long term. Thus, the analysis emphasizes the opinion that SR benefits companies with a better reputation and image and is more approved by markets even if they have a high impact, which instead reveals short and long-term benefits.

The risk and size variables both have significant negative effects. These results contribute to the relationship between SR and financial performance and show that profitability or the market is not affected negatively by sustainability initiatives based in an emerging market.

## 6. Implications and Further Research

The research has important implications for researchers who are interested in a more thorough exploration of the impact of SR on financial performance, as well as for practitioners who envision sustainability as the primary component of their business in the near future. The overall objective of this research is to understand the impact of SR on financial performance considering market and accounting measures and the industry of the firm in a developing country context. Although developing countries follow the developed ones in many issues, sustainability is very rapidly gaining importance in every market due to globalization and regulations. Future research can investigate this impact in different contexts. We, therefore, encourage future research to include other variables such as ownership structures and corporate governance indicators. Hence, future work might need to evaluate different firm-level or industry-level characteristics and the impact of SR on organizational outcomes. We also recommend that future research could consider a larger sample size or comparison of two or more different markets to have a larger perspective. Furthermore, future research can also consider the implications of new technologies such as blockchain on sustainability practices and their impact on firms' financial performance.

This research offers genuine insights for practitioners who envision sustainability as a primary component of their business. The positive relationships shown in the results express the importance of sustainability reporting and its positive impact on many perspectives, especially on ROA. Moreover, the analysis also highlights that SR helps companies build a better reputation and image which is more approved by all stakeholders. So, practitioners must consider the results of this study in their decision-making. Theoretically, the study contributes to the literature primarily via the addition of results from a developing market and the classification of firms into three industry categories.

## 7. Conclusions

The findings deduced from the study highlight that SR positively affects the accounting measures of the leading non-financial firms of a developing country in accordance with the findings of many previous studies, e.g., [22,46]. This positive effect is also well valued by investors, especially for higher impact firms in the short and long term. Moreover, the growth and the current ratio variables also has a positive impact on ROA and Tobin's Q as they are truly important for stakeholders. Accordingly, in this developing country context, stakeholders such as investors, shareholders, creditors, and debtors are recommended to increase their knowledge about SR and its importance in the business in order to make better investment choices. Furthermore, the risk and size variables display that profitability and the market are not affected negatively by SR. So, finally, we suggest that firms in developing countries focus more on sustainability reporting as a driver for better performance.

**Author Contributions:** Conceptualization, B.D., A.İ.K. and C.D.; methodology, B.D., A.İ.K. and C.D.; software, B.D., A.İ.K. and C.D.; validation, B.D., A.İ.K. and C.D.; formal analysis, B.D., A.İ.K. and C.D.; investigation, B.D., A.İ.K. and C.D.; resources, B.D., A.İ.K. and C.D.; data curation, B.D., A.İ.K. and C.D.; writing—original draft preparation, B.D., A.İ.K. and C.D.; writing—review and editing, B.D., A.İ.K. and C.D.; visualization, B.D., A.İ.K. and C.D.; supervision, B.D., A.İ.K. and C.D.; project administration, B.D., A.İ.K. and C.D.; funding acquisition, B.D., A.İ.K. and C.D. All authors have read and agreed to the published version of the manuscript.

**Funding:** This research received no external funding.

**Institutional Review Board Statement:** Not applicable.

**Informed Consent Statement:** Not applicable.

**Data Availability Statement:** Data can be retrieved from mentioned databases.

**Conflicts of Interest:** The authors declare no conflict of interest.

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
