# Peer review of "Nexus between Sustainability Reporting and Firm Performance: Considering Industry Groups, Accounting, and Market Measures"

_sustainability, doi:10.3390/su15075849_

Round 1
Reviewer 1 Report
The paper deals with a widely debated topic in the specialty literature, that of the link between sustainable reporting and company performance.
The research question is focused on the effect generated by sustainable reporting on performance indicators.
A wider argumentation of the studies carried out on this connection is necessary. It is also necessary to argue the use of the terms sustainable reporting (SR) with respect to ESG with the same meaning.
From a methodological point of view, the hypotheses must be highlighted, as well as robustness tests of the relationships tested through the proposed models.
The sample size is too small, potentially questioning the relevance of the conclusions associated with the analysis of the influence of the fields of activity.
A thorough analysis of the text is necessary. At the end of the Conclusions is the phrase "This section is not mandatory but can be added to the manuscript if the discussion is unusually long or complex"????
Best regards,
Author Response
Dear Reviewer,
First of all Thank you for your time and valuable comments and suggestions.
1- We tried to focus on SR-Performance relationship and Ä°n the introduction and Literature Review made the necessary changes to be more direct on this connection (also with the help of the table 1). In the beginnig of the literature review (L. 100-110). we also argued the SR and ESG terms.
2- We added explanation about the use of Robust St. Error L.224-232
3-For the sample size, we explained the data collection and we have 920 observations for 5 year period due to interruptions on SR activities of XU100 listed firms.
We also re-read and corrected whole text.
Best Regards
Reviewer 2 Report
Please find out the attachement report

Author Response
Dear Reviewer,
First of all Thank you for your time and valuable comments and suggestions.
1- Made the necessary changes in the abstract, introduction and literature review in order to be more direct.
2- Explained the source and collection of datain 3rd part Materials and methods also added more details
3- Highlighted the aim of the work in the text few timesenriched the literature review with your valuable suggestions (Thank you9
4- Rearranged all the parts of the text after results according to your comments and remarks
Thank you once again and Best Regards
Reviewer 3 Report
Thank you for allowing me to review this paper. However, I have shared my viewpoints.
The title of the article needs to be more precise.
The author(s) must highlight the fundamental objective in the abstract.
This finding, "Considering the environmental impact of companies, the results also reveal a positive relationship between 24 higher impact and short-term financial performance" is not clear.
The introduction needs to improve, mainly focusing more on the problem statement and research gap, which are the heart of academic research. So, the researchers need to focus seriously on these issues.
Table 1 resumes the recent studies about SR and firm performance. After the Table, you need to explain the Table to find out the research gap, which is very important.
The methodology part is OK.
Table 8. Summary of regression analysis for panel data. You have explained, but this part must be placed after the Table.
Discussion is one of the important parts of academic research, and it should be separated from the conclusion. It would be best if you focused on the "Discussion of the Findings". You need to discuss your own findings compared to the results of other researchers. If you find something completely new, you need to explain it adequately.
Before going to the conclusion, you must focus on the "Contribution or Implication" of your research. The contribution should be of two types which are theoretical contributions and practical or managerial contributions.
Regards.
Author Response
Dear Reviewer,
First of all Thank you for your time and valuable comments and suggestions.
1-We made the necessary changes according to your comments in the abstract and introduction e.g. tha gap L.83
2- Added exlanations for table 1 and made the place changes for better understanding of the paper
3- Rearrangeed the latest parts of the texts after the results according to your remarks.
Best Regards,
Round 2
Reviewer 1 Report
I agree with the changes made by the authors.
Best regards,
Reviewer 2 Report
no comments
Reviewer 3 Report
Dear authors
Thank you for the improvement of your manuscript. Best of luck for your manuscript.